# Development and Application of a Cleaved Amplified Polymorphic Sequence Marker (*Phyto*) Linked to the *Pc5.1* Locus Conferring Resistance to *Phytophthora capsici* in Pepper (*Capsicum annuum* L.)

**DOI:** 10.3390/plants12152757

**Published:** 2023-07-25

**Authors:** Giacomo Bongiorno, Annamaria Di Noia, Simona Ciancaleoni, Gianpiero Marconi, Vincenzo Cassibba, Emidio Albertini

**Affiliations:** 1Department of Agricultural, Food and Environmental Sciences, University of Perugia, 06121 Perugia, Italy; giacomo.bongiorno@studenti.unipg.it (G.B.); info@progeneseed.com (A.D.N.); simona.ciancaleoni@gmail.com (S.C.); gianpiero.marconi@unipg.it (G.M.); vince.cassibba@gmail.com (V.C.); 2Progene Seed s.s.a., 97019 Vittoria, Italy; 3Southern Seed s.r.l., 97019 Vittoria, Italy

**Keywords:** *Capsicum annuum*, CAPS markers, marker-assisted selection, soil-borne pathogens, *Phytophthora capsici*

## Abstract

*Phytophthora capsici* causes destructive disease in several crop species, including pepper (*Capsicum annuum* L.). Resistance in this species is physiologically and genetically complex due to many *P. capsici* virulence phenotypes and different QTLs and R genes among the identified resistance sources. Several primer pairs were designed to follow an SNP (G/A) within the *CA_011264* locus linked to the *Pc5.1* locus. All primer pairs were designed on DNA sequences derived from *CaDMR1*, a homoserine kinase (HSK), which is a gene candidate responsible for the major QTL on chromosome *P5* for resistance to *P. capsici*. A panel of 69 pepper genotypes from the Southern Seed germplasm collection was used to screen the primer pairs designed. Of these, two primers (*Phyto*_for_2 and *Phyto*_rev_2) surrounding the SNP proved successful in discriminating susceptible and resistant genotypes when combined with a restriction enzyme (*BtgI*). This new marker (called *Phyto*) worked as expected in all genotypes tested, proving to be an excellent candidate for marker-assisted selection in breeding programs aimed at introgressing the resistant locus into pure lines.

## 1. Introduction

Pepper (*Capsicum annuum* L., 2n = 2x = 24) is an economically important crop, cultivated almost all over the world for food (fresh vegetable or dried spice) production and non-food (cosmetics, pharmaceuticals, and pest management) purposes [1]. It represents the most popular and widespread pepper species among the five domesticated ones: *C. annuum*, *C. baccatum*, *C. chinense*, *C. frutescens*, and *C. pubescens* [2]. The cultivation of *C. chinense* and *C. frutescens* is typically limited to American, Asian, and African countries [1], while *C. baccatum* and *C. pubescens* are cultures mainly confined to Latin American countries such as Peru, Bolivia, Colombia, and Brazil [3]. From an economic point of view, the world’s annual commercial production of peppers is about 36 million tons, with over 16 million tons produced by China [4]. Cultivation in Italy occupies over 10,000 hectares, producing about 260,000 tons [4]. These numbers include different types of pepper such as jalapenos, cayenne, serrano, poblanos, chili peppers, bell peppers, and most other sweet peppers.

Like other horticultural species, pepper plant breeding is essentially based on different selection methods, and the choice of the best method or combination of them depends mainly on the type of inheritance (monogenic, oligogenic, or polygenic) from traits to be improved through breeding programs [5]. The strategy of pepper breeders is to develop a new F1 hybrid genotype (i.e., a pepper seed/plant that results from a cross-coupling of clearly different homozygous parental lines and that is characterized by an agronomically important condition known as heterosis or hybrid vigor) [6], with a higher genetic potential as the content of bioactive compounds, productivity, and resistance to different diseases [7]. The content of bioactive compounds is carefully taken into account in pepper breeding programs because all types of *C. annuum* peppers belonging to the genus *Capsicum* are an important source of phytochemicals with unique properties such as phenolic compounds, vitamins (C, E, and A), and capsaicinoids [8,9]. Among different phytochemicals, capsaicin is the main profitable bioactive sourced in the pepper fruit, with a vast number of traditional as well as pharmacological uses, which includes antioxidant, anticancer, anti-inflammatory, antischemic, antiarrhythmic, antiviral, antidiabetic, and antiulcer activity [8]. The specific properties of pepper fruits, coupled with the fact that pepper represents a profitable vegetable crop for food production, as it can ensure high production performance when grown in a controlled environment, make productivity another important goal of breeding programs. However, *C. annuum* is vulnerable to various pathogens, comprising viruses, bacteria, fungi, and nematodes [10,11], that cause enormous yield losses in pepper production and fruit quality. Thus, breeding for disease resistance undoubtedly occupies the most important position among the priorities of pepper breeders in producing new F1 hybrid cultivars.

Phytophthora blight has been identified as one of the most destructive diseases that significantly affects global pepper production [11]. The causal agent of Phytophthora blight is a soil-borne oomycete plant pathogen named *Phytophthora capsici* Leonian [12]. This pathogen can infect all parts of a pepper plant and be seed-borne, surviving in the soil for months [13,14,15,16]. The disease is recognizable for its distinct and typical syndromes, which include root rot, foliar blight, stem blight, and fruit blight [15,17,18,19]. The types of symptoms are mainly influenced by factors such as the host species, the point of infection, and environmental conditions [20]. Furthermore, plant maturity affects disease severity, with adult plants being more resistant than young plantlets [20]. Root rot is the most destructive of the disease syndromes, leading up to 100% yield loss under warm (25–28 °C) and humid environmental conditions [19]. The most evident symptom in pepper plants affected by root rot is wilting and death, even when the soil has enough moisture.

Moreover, in disease progression, the stem dries up and withers, dieback occurs, leaves defoliate, and the whole plant finally dies [21]. Although growers can manage fungal soil diseases by the adoption of strategies such as irrigation management, crop rotation, fungicidal applications, and host resistance [16,22,23], to date, no effective solutions have been found for adequate control of phytophthora blight in pepper cultivation [23]. This situation is probably due to some limitations that cannot be easily overcome, such as the survival of oospores in the soil for a long time [16,23,24,25], the presence of fungicide-resistant populations [16,26,27], and the lack of resistant cultivars with appealing horticultural characteristics [16,28].

Developing varieties of peppers resistant to *Phytophthora capsici* is an essential approach to control this disease. In pepper, decades of studies about the inheritance of the resistance to *P. capsici* have led to consider it as a physiologically and genetically complex trait due to the presence of several isolates of *P. capsici* that differ for the virulence phenotypes (where “*virulence phenotype*” supplement the term race to designate the virulence of the *P. capsici* isolates to the various host resistance genes) [29]. In fact, in *C. annuum*, the identified resistance to these different types is given by several QTLs [29,30,31,32,33,34,35,36,37,38,39,40,41,42,43,44,45] and R genes [38,41,43,43,45] not randomly distributed along the pepper genome.

Focusing on the main results archived during decades of research on the inheritance of the resistance to *P. capsici*, it follows that different authors have suggested the presence of important resistance factors on the P5 chromosome. Lefebvre and Palloix [30] were among the first to identify a major QTL on chromosome P5, which explained 41–55% of the phenotypic variance among 13 QTLs detected. Subsequently, Thabuis and colleagues [31] reported the existence of two closely linked QTLs, *rri5.1* and *rri5.2*, on chromosome P5. These QTLs were renamed *Phyto5.1* and *Phyto5.2*, respectively [32]. Ogundiwin and colleagues [33] detected a QTL, *Mr-5*, on chromosome P5 and stated that this QTL corresponded to *Phyto5.1* and *Phyto5.2*. However, Minamiyama and colleagues [34] reported that a direct correspondence of the QTL *Mr-5* to those described in previous studies is difficult to confirm because of the lack of common linkage markers. Quirin and collogues [35] detected *Phyto.5.2* as a major QTL for resistance to *P. capsici* mapped in the P5 chromosome and reported a sequence-characterized amplified region (SCAR) marker linked to this QTL. Sugita and colleagues [36] detected three QTLs (*Phyt-1*, *Phyt-2*, and *Phyt-3*) for resistance to Phytophthora blight and three markers (M10E3-6 AFLP, RP13-1 RAPD, and M9E3-11). Moreover, among the three QTLs, *Phyt-1* was identified in the same chromosomal region as other major QTLs previously found on chromosome P5, such as *Phyt.5.1* and *Phyt.5.2* [11]. Minamiyama and colleagues [34] identified an SSR marker (CAMS420) linked to a major QTL for resistance to *P. capsici*, which may correspond to *Phyto5.2*, and may possibly include *Phyto5.1*. In the same year, Bonnet and colleagues [37] reported the genetic map location of QTLs controlling the partial resistance to *P. capsici* and *P. parasitica* originating from the CM334 pepper genitor using the high-resolution map published by Barchi and colleagues [38]. In 2008, Kim and colleagues [39] developed different BAC-derived markers linked to *P. capsici* resistance, which were detected in P5 (one SNAP marker) and P9 (one Cleaved Amplified Polymorphic Sequence, CAPS, and two Simple Sequence Repeat, SSR, markers) chromosomes. Subsequently, Lee and colleagues [40] developed a molecular marker (M3-CAPS) closely related to the main QTL *Phyto.5.2* for resistance to *P. capsici* [40]. Truong and colleagues [41] developed a Random Amplified Polymorphic DNA (RAPD) marker (UBC553) and a Sequence Characterized Amplified Sequence (SCAR) marker (SA133_4), located in the linkage group P5 (chromosome P5), that correctly identified resistance or susceptibility to *P. capsici* in nine commercial pepper cultivars. By using anchor markers (i.e., molecular markers that serve as syntenic anchors to connecting genetic maps), Mallard and colleagues [42] detected three clustered QTLs, also located in chromosome P5 (*Pc5.1, Pc5.2, and Pc5.3*). Rehrig and colleagues [43] confirmed that the *Pc5.1* locus was the major resistance factor among these. The authors also found a candidate R gene (*DOWNY MILDEW RESISTANT 1, CaDMR1*) [43] closely linked to the *Pc5.1* locus. Liu and colleagues [44] developed a CAPS marker (NBS2_1-CAPS) derived from *Phyto*5NBS2_1 SNP located on chromosome P5, and HRM markers derived from three different SNPs (*Phyto*5NBS1, *Phyto*5NBS2_1, and *Phyto*5NBS2_1). Among these, the HRM marker based on *Phyto*5NBS1 SNP, and linked to an *NBS-LRR* candidate gene, showed high accuracy in predicting susceptible or resistant phenotypes to a low-virulence isolate (MY-1) of *P. capsici* [44]. Moreover, *Phyto*5NBS1 was mapped more distantly from the other two SNPs (*Phyto*5NBS2_1 and SNP *Phyto*5NBS2_1), which are linked to a different *NBS-LRR* candidate gene, positioned in a similar region of chromosome P5 where Rehrig and colleagues [43] found *CaDMR1*. Wang and colleagues mapped the race-specific resistance gene *CaPhyto*, to a 3.3 cM region between two SSR markers ZL6726 (29,097,205 bp) and ZL6970 (30,177,879 bp) on chromosome P5 [45]. Siddique and colleagues [46] confirmed the presence of the three major QTLs (*QTL5.1*, *QTL5.2*, and *QTL5.3*) in chromosome P5 by combining traditional QTL mapping and genome-wide association study (GWAS).

More recently, Du and colleagues [47] identified the *Snakin-1* (*SN1*) gene (*CA05g05250*) in the *Pc5.1* locus that controls the wide-spectrum resistance. Li and colleagues [48] identified five genes related to disease resistance in the *CQPc5.1* QTL region, which was earlier determined as locus *QTL5.1*. In addition, Zhang and colleagues [49] reported a GWAS-derived candidate region for the *P. capsici* resistance locus positioned at 23,844,243 to 25,526,786 bp on chromosome P5 of the Zunla-1 reference genome. Moreover, the authors stated that this region overlaps with the location of the *P. capsici* resistance loci *Pc5.1* and of the *QTL5.2* [49].

The knowledge of QTLs and R genes has encouraged the integration of marker-assisted selection (MAS) in conventional breeding programs. Moreover, several types of molecular markers linked to *P. capsici* resistance have been developed over the last two decades, such as simple sequence repeats (SSRs) [34,38,45,50], sequence-characterized amplified regions (SCARs) [32,41,45], cleaved amplified polymorphic sequences (CAPSs) [32,39,44,45], competitive allele-specific PCR (KASP) [49], and high-resolution melting markers (HRMs) [44,51,52]. However, some of these molecular markers are not widely used for MAS in practical breeding programs due to some use limitations.

Two rootstocks resistant to *P. capsici* have been registered: Pepperstorm from Fenix Seed (Catania, Italy) and Maqueda from Esasem (Verona, Italy). Using rootstocks as the source of resistance allows for overcoming some limitations in breeding novel pepper varieties resistant to Phytophthora blight. However, the number of *C. annuum* accessions (such as AC2258, PI201232, PI201234, Paladin, and CM334) showing resistance to *P. capsici* remains limited [15,19,53], with only Criollo de Morelos (CM334) that shows a very high degree of resistance to multiple races of this pathogen [28,54,55]. Considering the scenario described above, it is fundamental to identify as many genetic resistance resources as possible and to develop new molecular markers to improve selection efficiency and allow rapid screening of individuals carrying the resistance genes during breeding. Rehrig and colleagues found that a sequence (*CA_011264*) derived from *CaDMR1,* a gene encoding for a homoserine kinase (HSK), contained an SNP that co-segregated with the resistance to *P. capsici* in pepper (CM334/Early Jalapeño alleles, for resistance and susceptibility, respectively) [43].

Since cleaved amplified polymorphic sequences (CAPSs) represent a class of molecular markers that are very helpful in MAS [56,57], in this work, we used the previously published information about the CA_01126_SNP sequence [43] to develop and validate a codominant CAPS marker, that we named *Phyto*, to be used in genotype selection to simplify or eliminate the need for phenotypic screening during breeding programs.

## 2. Results

### 2.1. Development of CAPS Marker

Rehrig and colleagues [43] produced a high-density map of 3887 markers for a segregating population derived from a cross between an accession (CM334) highly resistant to *P. capsici* and a susceptible one (Early Jalapeno). Among these markers, the authors identified two SNPs, CA_011264 and CA_004482, located within the gene *CaDMR1*, that correlated highly with *P. capsici* resistance, pinpointing *CaDMR1* as a strong candidate for a *P. capsici* resistance in the root-rot-resistant pepper lines studied. Moreover, a strong correlation between *CA_011264* locus and resistance to *P. capsici* was later confirmed in three separate pepper populations [43].

Therefore, in this work, we used the previously characterized CA_011264 SNP (G/A) information to develop and validate a CAPS marker to be used in MAS programs. To this end, we employed the 2159 bp DNA sequence surrounding the *CA_011264* locus to design several forward and reverse primers (Table 1).

These primers were tested on nine different pepper pure lines with known resistance/susceptibility behavior and showed to be highly efficient in producing amplicons (data not shown). Amplicons were sequenced and confirmed the presence of the SNP (Figure 1 and Appendix A). We also determined that the SNP caused the loss of a *BtgI* restriction site in the resistant background (Figure 1 and Appendix A).

Some primer pairs, for example, *Phyto*_for_1/*Phyto*_rev_1, produced amplicons of small size that, even if helpful in determining the presence of resistance and/or susceptible allele by Sanger sequencing, when used for CAPS analysis, produce small-digested fragments uneasily distinguishable after running in agarose gel.

Although all the other primer pairs could be employed for a valid amplification/digestion of the desired locus, we decided to use the *Phyto*_for_2 and *Phyto*_rev_2 primer pair since they amplify an amplicon of 477 bp that, after *BtgI* digestion, allow to easily distinguish the resistant (an undigested band of 477 bp) and the susceptible (two bands of 343 and 138 bp, because of 4 bp protruding sequence) genotypes (Figure 2).

As a result, we determined that the combination of *Phyto*_for_2/*Phyto*_rev_2 with *BtgI* enzyme represents a valid CAPS marker, hereafter named *Phyto*, for discriminating genotypes that differ in their resistance to *P. capsici*.

### 2.2. CAPS Validation

To evaluate the usefulness and reproducibility of the marker for its application in MAS breeding, the *Phyto* CAPS marker was tested on the entire set of 69 pepper genotypes with different genetic backgrounds and a known resistance status. As expected, after digestion, all homozygous resistant lines showed the 477 bp band, corresponding to the resistant allele (Figure 2, Appendix A), while all homozygous susceptible genotypes showed two fragments (343 and 138 bp). The heterozygous genotypes showed three fragments (477, 343, and 138 bp) corresponding to the undigested resistant and the digested susceptible alleles (Figure 2, Appendix A).

## 3. Discussion

The production of *Capsicum annuum* L., one of the most economically valuable horticultural crops, is hampered by several diseases such as anthracnose, powdery mildew, phytophthora root rot, cucumber mosaic virus, tomato spotted wilt virus, bacterial spot, and bacterial wilt. Therefore, breeding peppers to introduce resistance to multiple diseases is highly sought after [58]. In fact, developing such varieties could reduce the frequency of resistance breakdown, which would result in mitigating yield and quality loss due to the infection of pathogens. This goal can be achieved by the rapid introgression of disease resistance genes aided using marker-assisted selection [59,60]. Over the years, several markers have been developed and tested in pepper [61].

In the present study, we have focused on one of the most destructive pathogens impacting pepper production, *Phytophthora capsici*. In pepper, it is responsible for significant losses in yield and quality of fruits. The plant syndromes disease caused by *P. capsici* can be controlled through biological, chemical, and mechanical ways [10,16,22]. Among these, disease control through chemicals is the most widespread and popular strategy adopted by the farming community [10]. Fungal diseases in pepper are usually controlled by fungicides such as mefenoxam, fluopicolide, oxathiapiprolin, dimethomorph, mandipropamid, and cyazofamid [27] or by cultural practices like crop rotation and irrigation management [16,22]. However, managing diseases through chemical compounds is becoming difficult due to increased resistance to pathogens [62], and it can also generate several environmental problems. Moreover, as new virulent pathogen races continue to emerge within the context of climate change [63], applying pesticides and fungicides against pests should be strongly discouraged. To this end, integrated disease management strategies such as host plant resistance, cultivation practices, and biological control should be combined to control disease caused by *P. capsici*.

Developing pepper cultivars with renewed resistance to *Phytophthora* races is an efficient and sustainable strategy to control phytophthora blight disease [20], an alternative to the widespread use of fungicides for phytophthora mitigation. Fundamentally, breeding approaches to developing resistant pepper cultivars are based on the evaluation of germplasm for possible sources of resistance, introgression of the source of resistance into an elite background, and deployment of the F1 hybrid cultivar into other existing disease management strategies. In addition, pepper germplasm for resistance is evaluated using selection criteria that must be strict to ensure only the transfer of the most resistant material, as there might be many factors that do not correspond to the priorities of the breeding activity. To this end, the use of molecular markers through MAS provides valuable solutions to overcome some of the problems faced by classical phenotypic screening approaches in plant breeding programs [56,59,60,64]. Molecular markers can be used to tag QTLs and R genes linked to resistance traits, and further, their use can improve efficiency for selecting germplasm sources of resistance [65]. In addition, marker-assisted selection represents a valuable tool that, when used at an early stage of pepper plant development with multiple molecular markers, can lead to simultaneously selecting a segregating population for one or more resistance-trait-related loci.

Regarding the *P. capsici* resistance in pepper, the knowledge of mechanisms that control this complex type of inherited resistance played a key role in the identification and introduction of molecular markers through MAS in breeding programs, as, for instance, *Phyto*5NBS1 (HRM), ZL6726 (SSR), NBS1-CAPS and CaDMR1-dCAPS. These are molecular markers identified in a region of chromosome P5 that is recognized as the carrier of important resistance factors, such as QTLs loci (*Pc5.1* and QTL*5.2*) and R genes (*CaDMR1*, *NBS-LRR*, and *CaPhyto*). Developed by Liu and colleagues [44], *Phyto*5NBS1 is an HRM marker derived from an SNP (*Phyto*5NBS1) that shows high accuracy in predicting susceptible or resistant phenotypes against a low-virulence isolate (MY-1) of *P. capsici* [44]. Like the *Phyto*5NBS1 (HRM) marker, NBS1-CAPS is a CAPS marker based on *Phyto*5NBS1 SNP. Developed by Wang and colleagues [45], this marker has been mapped to a 3.3 cM chromosomal interval between markers CaDMR1-dCAPS and ZL6726, close to the *CaPhyto* gene. However, NBS1-CAPS has been utilized for the construction of a genetic linkage map [45], and, therefore, its effectiveness in detecting disease phenotypes has not been determined. CaDMR1-dCAPS [45] is another potentially useful molecular marker derived from SNP *Ca_011264* linked to the *Pc5.1* locus [43]. However, as with NBS1-CAPS, there is no information available on the efficacy of this d-CAPS marker in detecting disease phenotypes and its practical use in pepper breeding programs. Instead, ZL6726 is an SSR marker that has been validated to be a reliable marker for the selection of resistance phenotypes to *P. capsici* race 2 [45].

Currently, *Phyto*5NBS1 (HRM) [44] and ZL6726 (SSR) [45] represent two available molecular markers linked to a major resistance factor in the P5 chromosome for marker-assisted selection of *Phytophthora capsici* resistance. However, the applicability of these markers sometimes has limitations [44] because their accuracy can show significant variations when used with different pepper germplasms presenting disease-specific phenotypes [46]. In fact, the genotypes of these two reported molecular markers perform differently when matched with resistance phenotypes tested with different degrees of virulence (*Phyto*5NBS1) [44] and when applied to relatively small groups of pepper lines and commercial cultivars (ZL6726) [45]. This phenotype–genotype mismatch represents an important bottleneck that limits the efficiency of MAS [21], making breeding for *P. capsici* resistance challenging. As a result, there is still a need to discover and validate new molecular markers linked to the R and/or QTL genes with respect to *P. capsici* in pepper gene pools.

In this study, we employed the sequence surrounding the *CA_011264* locus in *CaDMR1* located on the P*5* chromosome [43] and highly linked to the *Pc5.1* locus. This sequence includes an SNP within the *CA_011264* locus linked to *P. capsici* resistance. The Ca_011264 SNP is a type of genetic variation, based on nucleotide transition (G/A), that we discovered can be used in rapid and relatively low-cost assays, developing a CAPS marker based on the PCR method to select genotypes with broad-spectrum resistance to *P. capsici.* In fact, as demonstrated by Rehrig and colleagues [43], the resistant allele in the *Ca_011264* locus, which is conserved in CM334 and other resistant accessions, showed a strong correlation with resistant phenotypes in different pepper populations. Moreover, this has been found in a QTL region validated for conferring resistance to multiple isolates of *P. capsici*, including five of the most aggressive *P. capsici* isolates from those that originated from New England, New Mexico, and Mexico. Therefore, we used this information previously published by Rehrig and colleagues [43] to develop an SNP-based CAPS marker and validate its applicability to rapidly distinguishing pepper plants resistant/susceptible to *P. capsici* at the seedlings stage to be used in breeding programs.

An essential step in the process of marker validation is defined by the testing of the developed marker for its effectiveness in discriminating phenotypes in different genetic backgrounds [66]. To validate the functionality of the *Phyto* CAPS marker, we tested it on a panel of 69 pepper genotypes belonging to the Southern Seed germplasm collection, well characterized for resistance/susceptibility to *Phytophthora capsici* and with a different genetic background. In accordance with the phenotypic data shared by the owner of the germplasm, all genotypes that resulted homozygous and heterozygous for the *Phyto* R allele also showed a high level of resistance against the *P. capsici* isolates most spread in the Southern Italy area, whereas different levels of susceptibility were observed in genotypes that resulted homozygous for the susceptible allele of the *Ca_011264* locus. These differences might be related to other resistance factors linked to *P. capsici* resistance, which can be distributed along pepper chromosomes.

*Phyto* CAPS results to be an excellent candidate for marker-assisted selection in breeding programs aimed to speed up the efficiency in screening resistant/susceptible plants to *P. capsici*. In contrast to the results observed using the before-mentioned molecular markers, *Phyto* is highly accurate in detecting resistant and susceptible pepper phenotypes. Furthermore, being based on an SNP located within the *DOWNY MILDEW RESISTANT* 1 gene, which was pinpointed as a candidate gene for resistance to multiple *Phytophthora capsici* isolates, it represents a valuable tool for the rapid identification of this gene and the development of new pepper lines with a wider spectrum of resistance. In addition, co-segregating with QTL *Pc5.1*, it is also a suitable tool in breeding programs aimed at pyramiding the *Pc5.1* locus with different genes traits of interest.

Starting from F2 generation, *the Phyto* marker can be used with other molecular markers to identify superior allele combinations rapidly. Instead, after the selection of parents for the development of new segregant populations, if the selected parent lines are not well characterized, the use of *Phyto* is not recommended because only a small number of segregant lines should be cut off from breeding populations during the first generations. No lineages should be screened for disease resistance or more complex traits up to F4 or F5 generations. Selection pressure can reach a significative intensification at F4 generation and/or in more advanced generations when selection procedures for accumulating many minor genes/QTLs to several traits simultaneously become more efficient than in early generations. In these generations, the *Phyto* marker can be used at the early stage of pepper plant development to rapidly discard pepper lines that are susceptible to *P. capsici*. In addition, in the availability of molecular markers in perfect linkage with other R genes or QLTs, *Phyto* can be used in breeding programs aimed at pyramiding different gene traits of interest. Finally, *Phyto* allows a rapid evaluation of the allelic status in the *Ca_011264* locus, either before or after cross-coupling, aimed at developing new hybrid F1 pepper cultivars.

## 4. Materials and Methods

### 4.1. Plant Materials and DNA Extraction

A total of sixty-nine different *Capsicum annuum* genotypes belonging to Southern Seed’s germplasm collection, genetically unrelated and with known resistance/susceptibility to *Phytophthora capsici,* were used. This germplasm collection comprised 32 F1 genotypes (all resistant) and 37 pure lines (14 resistant and 23 susceptible). The plants were grown in plastic pots in a greenhouse. Fresh young leaves of individuals were collected (separately from resistant, susceptible, and F1 individuals) and preserved at −70 °C until DNA extraction. Genomic DNA was extracted with GenEluteTM Plant Genomic DNA Kit (Sigma Aldrich, St. Louis, MO, USA) according to the manufacturer’s instructions. The quality and quantity of DNA were measured using NanoDrop™ 2000c Spectrophotometer (Thermo Fisher, Waltham, MA, USA).

### 4.2. Marker Development

Restriction sites in the *CA_011264* locus were identified using NEBcutter [67]. Primer 3 was employed to design CAPS markers. Flanking sequences comprising the CA_011264 SNP were extracted from the sequence CA00g97170 sourced in the “*Capsicum annuum* cv CM334 Genome CDS (Release 1.55)” BLAST database, available through the Sol Genomics Network (SGN) web portal [68] and used for primers design. PCR mixture (50 μL) used to perform each analysis contained DreamTaq™ Hot Start DNA Polymerase (Thermo Fisher, Waltham, MA, USA), 10 pmole·μL^−1^ forward and reverse primers, distilled water, and 25 ng·μL^−1^ DNA. PCR was performed using the *SimpliAmp* Thermal Cycler (Thermo Fisher, Waltham, MA, USA) under the following conditions: initial denaturation at 94 °C for 5 min, followed by 35 cycles of denaturation at 94 °C for 30 s, annealing at 62 °C for 30 s, and extension at 72 °C for 1 min with a final extension at 72 °C for 5 min. PCR products were digested according to the protocol with the *BtgI* restriction enzyme (R0608S, NEB). The digested PCR amplicon was observed using 2.2% agarose gel electrophoresis with a UV light system.

## 5. Conclusions

Being user-friendly, reliable, and cheap, the marker developed in this study is a valuable tool for pepper breeding programs. Moreover, if coupled with other markers, it would help in breeding for genotypes carrying multiple resistances, a requirement more and more needed. *Phyto* is a co-dominant and specific locus marker that allows distinguishing resistant and susceptible homozygotes, as well as heterozygotes. In addition, it is a valuable tool that can reduce the cost of integrating MAS into breeding programs, as the equipment required for its use makes genotype-based selection relatively cheap.

## Figures and Tables

**Figure 1 plants-12-02757-f001:**
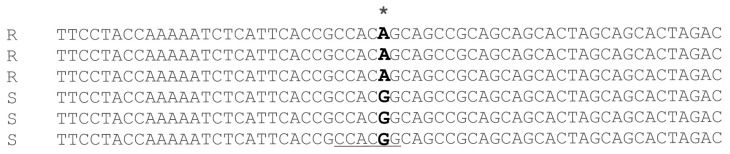
Multiple sequence alignment showing the SNP between *P. capsici* resistant (R) and susceptible (S) homozygous lines. The SNP is bolted, and its position is marked with an asterisk (*); underlined sequence shows the *BtgI* restriction site.

**Figure 2 plants-12-02757-f002:**
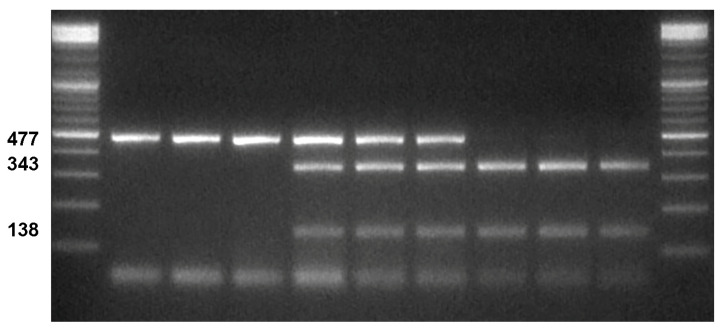
Gel electrophoresis of the digestion products of *Phyto*. From left to right: ladder; three homozygote resistant genotypes (showing a single band of 477 bp); three heterozygote genotypes (showing the undigested band of 477 bp and the 343 and 138 bp bands, produced by the *BtgI* digestion of the 477 bp band); three susceptible homozygote genotypes (showing only the 343 and 138 bp bands); ladder.

**Table 1 plants-12-02757-t001:** Primer set designed for PCR amplification of DNA sequence surrounding Ca_011264 locus.

Primer Set	Sequence (5′-3′)	Product Size (bp)
*Phyto*_for_1	AGCTGATCAACACTCAATTTCCT	98
*Phyto*_rev_1	CCGTTGGGTAGTGGACTTGG	
*Phyto*_for_1	AGCTGATCAACACTCAATTTCCT	182
*Phyto*_rev_2	TGTGCTGGAATTGCTGCTTT	
*Phyto*_for_1	AGCTGATCAACACTCAATTTCCT	373
*Phyto*_rev_3	CTTCCGTCAAATCCTTCGCC	
*Phyto*_for_2	CCTCGAATCCCCTTGCAGTA	393
*Phyto*_rev_1	CCGTTGGGTAGTGGACTTGG	
*Phyto*_for_2	CCTCGAATCCCCTTGCAGTA	477
*Phyto*_rev_2	TGTGCTGGAATTGCTGCTTT	
*Phyto*_for_2	CCTCGAATCCCCTTGCAGTA	668
*Phyto*_rev_3	CTTCCGTCAAATCCTTCGCC	
*Phyto*_for_3	ATCTCACAAGTGGGGTCTGG	923
*Phyto*_rev_1	CCGTTGGGTAGTGGACTTGG	
*Phyto*_for_3	ATCTCACAAGTGGGGTCTGG	1007
*Phyto*_rev_2	TGTGCTGGAATTGCTGCTTT	
*Phyto*_for_3	ATCTCACAAGTGGGGTCTGG	1198
*Phyto*_rev_3	CTTCCGTCAAATCCTTCGCC	

## Data Availability

Not applicable.

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
