# Peer review of "Development and Application of a Cleaved Amplified Polymorphic Sequence Marker (Phyto) Linked to the Pc5.1 Locus Conferring Resistance to Phytophthora capsici in Pepper (Capsicum annuum L.)"

_plants, 2023, doi:10.3390/plants12152757_

Round 1

Reviewer 1 Report

The manuscript submitted by Bongiorno and colleagues describes the development and testing of a new CAPS marker against Phytophtora, the most serious pathogen of peppergrass (Capsicum annum). The study, given the economic importance of the crop, is very interesting and, in my opinion, should be of broad interest.

I make some suggestions to improve the quality of the manuscript: 

The only important flaw of the manuscript is that the discussion chapter does not compare the results with previous results, it is almost an extension of the introduction, so I suggest rewriting the chapter along the following questions:

Why is the marker you developed better than the previous ones? (Is it more cost-effective?)

What are the advantages/disadvantages of the developed marker compared to the previous ones?

At which point in pepper breeding is the use of the marker recommended?(To my knowledge, most of the pepper varieties grown today are F1 hybrids. In the introduction I suggest at least briefly mentioning this and the importance of the heterosis effect in pepper.)

Pyramiding genes is of great importance for resistance breeding in horticultural crops. How can this be exploited in pepper breeding and how can the newly developed marker help?

In order not to waste a lot of work in the discussion section, I suggest that the points made there should be incorporated in the introduction.

The conclusions section should be short and to the point. I recommend keeping only the last paragraph.

Minor note: In Figure 2, the 204 bp long fragment is not visible. Although the 3 genotypes are still clearly visible and separable, I suggest replacing the gel image if possible.

Author Response

Please find below a detailed list of answers (in bold) to each comment raised by the reviewers.

Reviewer 1

1 - Why is the marker you developed better than the previous ones? (Is it more cost-effective?)

Being based on an SNP located within the gene DOWNY MILDEW RESISTANT 1 (CaDMR1), which was pinpointed as a candidate gene for resistance to multiple Phytophthora capsici isolates, it represents a valuable tool for the rapid identification of this gene and the development of new pepper lines with a wider spectrum of resistance.

Moreover, in contrast to the previously published results (mainly the markers were used for mapping and, at least as far as we know, never tested on a wide range of genetic backgrounds), Phyto exhibits a high accuracy in detecting resistant and susceptible pepper phenotypes in an easy and fast way (as all CAPS do). This marker can be used without the need for DNA isolation (employing a direct PCR kit) and gives results in a few hours speeding up the work of breeders dealing with hundreds of plants in segregating (Pedigree/SSD) or backcrossing breeding populations.

2 - What are the advantages/disadvantages of the developed marker compared to the previous ones?

See the reply to question 1

Moreover, since Phyto co-segregate with QTL Pc5.1, it is also a suitable tool in breeding programs aimed at pyramiding Pc5.1 locus with different genes traits of interest.

3- At which point in pepper breeding is the use of the marker recommended?(To my knowledge, most of the pepper varieties grown today are F1 hybrids. In the introduction I suggest at least briefly mentioning this and the importance of the heterosis effect in pepper.)

As mentioned in the answer to question 1, this marker can be used without the need for DNA isolation (employing a direct PCR kit) and gives results in a few hours speeding up the work of breeders dealing with hundreds of plants in segregating (Pedigree/SSD) or backcrossing breeding populations. Therefore, it must be used in the setup of one of the pure lines that will be used to produce the F1 hybrids. As for the hybrid vigor, in the introduction, we have rearranged the text as follows:

“The strategy of pepper breeders is to develop a new F1 hybrid genotype (i.e., a pepper seed/plant that results from a cross-coupling of clearly different homozygous parental lines and that typically is characterized by an agronomically important condition known as heterosis or hybrid vigor), with a higher genetic potential as the content of bioactive compounds, productivity, and resistance to different diseases [6].”

4- Pyramiding genes is of great importance for resistance breeding in horticultural crops. How can this be exploited in pepper breeding and how can the newly developed marker help?

Following the reviewer suggestion we have rearranger the discussion as follow:

“Starting from F2 generation, Phyto marker can be used in combination with other molecular markers, for the rapid identification of superior allele combinations. Instead, after the selection of parents for the development of new segregant populations, if the selected parent lines are not well characterized, it is not recommending the use of Phyto because only a small number of segregant lines should be cut off from breeding populations during the first generations and no lineages should be screened for disease resistance or more complex traits up to F4 or F5 generations. Selection pressure can reach a significative intensification at F4 generation, and/or in more advanced generations, when selection procedures for the accumulation of many minor genes/QTLs to several trait simultaneously become more efficient than in early generations. In these generation the Phyto marker can be used at the early stage of pepper plant development to rapidly discard pepper lines that result susceptible to P. capsici. In addition, in the availability of molecular markers in perfect linkage with other R genes or QLTs, Phyto can be used in breeding programs aimed at pyramiding different genes traits of interest. Finally, Phyto allows to a rapid evaluation of the allelic status in Ca_011264 locus, either before or after cross-coupling aimed at developing new hybrid F1 pepper cultivars.”

5 - In order not to waste a lot of work in the discussion section, I suggest that the points made there should be incorporated in the introduction.

We have moved several sections to the introduction.

6 - The conclusions section should be short and to the point. I recommend keeping only the last paragraph.

Done

7 - Minor note: In Figure 2, the 204 bp long fragment is not visible. Although the 3 genotypes are still clearly visible and separable, I suggest replacing the gel image if possible.

Done

Reviewer 2 Report

In this work, the authors developed a SNP-based CAPS marker (phyto) linked to Pc5.1 gene conferring resistance to Phytophthora capsici in pepper. However, there are some deficiencies.

1.     Authors declared that “there is no useful marker for MAS of phytophthora resistance because the accuracy of P. capsici resistance-related markers shows significant variation when used with different pepper genotypes that exhibit specifi disease phenotypes” (Line 113). So the advantage of marker (phyto) should be fully demonstrated in the manuscript. But here I can’t see the details. The authors should list the 69 lines and their genotyping, as well as the correlation between the genotypes and phenotypes.

2.     For 2.2 section, the author should present the figure of detection of 69 lines rather than use Fig 2, which only exhibited the efficiency of marker (phyto) distinguishing homozygous R / S lines and heterozygote lines.

3.     In my opinion, it is confused that two bands of 343 bp and 204 bp can be produced by digesting with an amplicon of 477 bp. 343+204=547? It greatly shakes my trust to the data. Since the products had been sequenced, please show the whole sequence of amplicon.

4.     The introduction and discussion could be more focused. Most of them were not really tightly around Pc5.1 & its molecular marker and suggested to be deleted. Sepcially, the authors should introduce or discuss the previously developed molecular markers linked to Pc5.1 and compared them with this marker (phyto).

Author Response

Reviewer 2

  1. Authors declared that “there is no useful marker for MAS of phytophthora resistance because the accuracy of P. capsici resistance-related markers shows significant variation when used with different pepper genotypes that exhibit specific disease phenotypes” (Line 113). So, the advantage of marker (phyto) should be fully demonstrated in the manuscript. But here, I can’t see the details. The authors should list the 69 lines and their genotyping, as well as the correlation between the genotypes and phenotypes.

As reported in the response to reviewer 1, “…in contrast to the previously published results (mainly the markers were used for mapping and, at least as far as we know, never tested on a wide range of genetic backgrounds), Phytoexhibits a high accuracy in detecting resistant and susceptible pepper phenotypes in an easy and fast way (as all CAPS do). This marker can be used without the need for DNA isolation (employing a direct PCR kit) and gives results in a few hours speeding up the work of breeders dealing with hundreds of plants in segregating (Pedigree/SSD) or backcrossing breeding populations.”

As regards the list of genotypes, these 69 genotypes that we used for testing the reliability of the marker belong to a private company. For each sample, they provided us the tolerance status (resistant vs. susceptible), and the genotype (pure line or F1 hybrid). For each sample they sent us the leaves.

If the reviewer believes it will be useful, we could provide a supplementary table of all genotypes (coded anonymously since they are breeding materials not commercial). Only few samples belong to commercially available lines. We can provide the resistance vs. susceptible behavior and the banding pattern obtained for each genotype.

We did not produce such a table because we believed it did not give any additional information to what was already written in the manuscript.

But we leave the editor the decision, and in case, we will provide such a supplementary table.

  1. For 2.2 section, the author should present the figure of detection of 69 lines rather than use Fig 2, which only exhibited the efficiency of marker (phyto) distinguishing homozygous R / S lines and heterozygote lines.

We decided to use a summary figure clearly showing the band's status. Providing a figure with 69 genotypes would result in several figures (our high-quality gel system allows us to run 13 samples per time, and if we consider that 2 lanes are used for ladders, each gel has room only for 10 samples. Therefore, we would need to show 7 figures, which we believe will not add any information to what is already reported in the manuscript.

  1. In my opinion, it is confused that two bands of 343 bp and 204 bp can be produced by digesting with an amplicon of 477 bp. 343+204=547? It greatly shakes my trust to the data. Since the products had been sequenced, please show the whole sequence of amplicon.

The reviewer was right. There was a problem with the editing of the manuscript, and we messed up with the size written. Still, as evident in the previous Figure 2 (now also replaced by a new Figure 2), the smaller fragment is below the 200 bp (so the data are reliable, we just had problems during the editing of the final version of the manuscript). We, therefore, thank the reviewer for having pointed it out.

As mentioned above, we have also produced a new Figure 2 and a Supplementary Figure 1 with the entire sequence alignment to meet the reviewer's requirements.

Therefore, the right dimensions of the digested fragments are 343 and 138. The sum of the two digested fragments does not make 477 but 481 because the enzyme is a sticky-end enzyme, and there are 4 bp extruding in both fragments.

  1. The introduction and discussion could be more focused. Most of them were not tightly around Pc5.1 & its molecular marker and suggested to be deleted. Specially, the authors should introduce or discuss the previously developed molecular markers linked to Pc5.1 and compared them with this marker (phyto).

We have modified the Introduction and discussion accordingly to the reviewer’s suggestions.

Reviewer 3 Report

This manuscript has new information about CAPS marker linked to the ressistance to Phytiphthora capsici in pepper for MAS. I think it is sufficient to publish.

Author Response

we would like to thank the reviewers that have helped us in improving the manuscript

Round 2

Reviewer 1 Report

Thank you for revising your manuscript. In my opinion,  the quality of the manuscript have been significantly improved, and is ready for publication.

Author Response

We would like to thank the reviewer for the positive feedback

Reviewer 2 Report

It seemed that the MS got much better, especially after correcting the wrong data and replacing the image. However, in my opinion, it could be improved.

1. I’m not so satisfied with the response for what is the advantage of Phyto CAPS maker. Although the authors add some information in the discussion section (line 323-334), I persist in the details should be provide as a strong proof in “2.2” section, either table or figure. It is the highlight of this study to demonstrate high accuracy of Phyto CAPS maker by validating using 69 genotypes with different genetic background.

2. Why only choose “for_2 and rev_2 prier pair” as the primers of Phyto CAPS maker? Line 206-209, the authors expounded the disadvantage of the product with small size. However, for other primer sets, the authors did not mention. Furthermore, the effects could not be compared due to unavailable information of other primer sets (line 198). Thus, the reason choosing the primer pair of Phyto_for_2 and Phyto_rev_2 as the final CAPS marker was not well explained.

Minor comments:

1.  Supplementary Information should be added in the end of the text.

2. There were several mistaken in writing. For examples, Marker-assisted selection (MAS) and downy mildew resistant 1 (CaDMR1) should be emerged in the first time and then abbreviated; Mt/million tons; chromosome P5/ chromosome 5.

3.  Line 194, “containing” could be “surrounding”

4.  Line 373, “4.1” should be “4.2”

5. Line 379, “DNA Polymerase (2X)” could be some reagent such as “MasterMix (2X)”.

Author Response

Please find below the response to each question:

1 - I’m not so satisfied with the response for what is the advantage of Phyto CAPS maker. Although the authors add some information in the discussion section (line 323-334), I persist in the details should be provide as a strong proof in “2.2” section, either table or figure. It is the highlight of this study to demonstrate high accuracy of Phyto CAPS maker by validating using 69 genotypes with different genetic background

We produced a Table as supplementary Table 1.

2 - Why only choose “for_2 and rev_2 prier pair” as the primers of Phyto CAPS maker? Line 206-209, the authors expounded the disadvantage of the product with small size. However, for other primer sets, the authors did not mention. Furthermore, the effects could not be compared due to unavailable information of other primer sets (line 198). Thus, the reason choosing the primer pair of Phyto_for_2 and Phyto_rev_2 as the final CAPS marker was not well explained.

Since all primer pairs worked, we have chosen one that gave a robust amplification product and whose digested fragments were easily scorable. It was just a choice. We do not feel like explaining it much more in the manuscript since we already state that: Although all the other primer pairs could be employed for a valid amplification/digestion of the desired locus, we decided to use the Phyto_for_2 and Phyto_rev_2 primer pair…..

Minor comments:

  1. Supplementary Information should be added in the end of the text. DONE
  2. There were several mistaken in writing. For examples, Marker-assisted selection (MAS) and downy mildew resistant 1 (CaDMR1) should be emerged in the first time and then abbreviated; Mt/million tons; chromosome P5/ chromosome 5. DONE
  3. Line 194, “containing” could be “surrounding” DONE
  4. Line 373, “4.1” should be “4.2” DONE
  5. Line 379, “DNA Polymerase (2X)” could be some reagent such as “MasterMix (2X)”. REMOVED 2X since this is the name of the KIT